# An Automatic Method for Elbow Joint Recognition, Segmentation and Reconstruction

**DOI:** 10.3390/s24134330

**Published:** 2024-07-03

**Authors:** Ying Cui, Shangwei Ji, Yejun Zha, Xinhua Zhou, Yichuan Zhang, Tianfeng Zhou

**Affiliations:** 1School of Mechanical Engineering, Beijing Institute of Technology, Beijing 100081, China; bitcuiying@163.com (Y.C.); zyc@bit.edu.cn (Y.Z.); 2School of Medical Technology, Beijing Institute of Technology, Beijing 100081, China; 3Department of Orthopedic Trauma, Beijing Jishuitan Hospital, Beijing 100035, China; ddv950315@126.com (S.J.); yijune23@126.com (Y.Z.); 4Department of Orthopedics, Beijing Jishuitan Hospital, Beijing 100035, China; bjzhouxinhua@sina.com

**Keywords:** elbow computerized tomography (CT) image, elbow MedSAM, bone recognition, medical image segmentation, three-dimensional (3D) reconstruction

## Abstract

Elbow computerized tomography (CT) scans have been widely applied for describing elbow morphology. To enhance the objectivity and efficiency of clinical diagnosis, an automatic method to recognize, segment, and reconstruct elbow joint bones is proposed in this study. The method involves three steps: initially, the humerus, ulna, and radius are automatically recognized based on the anatomical features of the elbow joint, and the prompt boxes are generated. Subsequently, elbow MedSAM is obtained through transfer learning, which accurately segments the CT images by integrating the prompt boxes. After that, hole-filling and object reclassification steps are executed to refine the mask. Finally, three-dimensional (3D) reconstruction is conducted seamlessly using the marching cube algorithm. To validate the reliability and accuracy of the method, the images were compared to the masks labeled by senior surgeons. Quantitative evaluation of segmentation results revealed median intersection over union (IoU) values of 0.963, 0.959, and 0.950 for the humerus, ulna, and radius, respectively. Additionally, the reconstructed surface errors were measured at 1.127, 1.523, and 2.062 mm, respectively. Consequently, the automatic elbow reconstruction method demonstrates promising capabilities in clinical diagnosis, preoperative planning, and intraoperative navigation for elbow joint diseases.

## 1. Introduction

Computerized tomography (CT) scans are widely applied in the medical imaging field, providing rapid and accessible imaging for the musculoskeletal system [1]. Moreover, CT scans have significant advantages in three-dimensional (3D) representation, allowing for the detailed description and quantification of complex anatomical regions. They are commonly employed in elbow examinations for various elbow pathologies [2,3,4], including acute elbow trauma [5,6], fracture-dislocation [7], degenerative changes [8], and elbow osteoarthritis [9,10]. Furthermore, elbow CT examinations assist in personalized implant design, precise alignment [11], elbow joint kinematics, dynamic analysis [12], and personalized treatment [13,14,15], thereby improving surgical accuracy and patient safety. However, it is essential for senior surgeons to perform bone reconstruction before diagnosing based on CT images.

In clinical practice, the reconstruction of bone surface models is often cumbersome. Initially, bone reconstruction was performed using engineering modeling software [16], including AutoCAD and SolidWorks. However, with recent advances in computer-aided medicine, medical analysis software has made bone reconstruction more scientific and reasonable. Willing et al. [12] generated 3D elbow joint models using threshold segmentation and performed wrapped and smoothed operations based on Mimics. Bizzotto et al. [17] created articular models by uploading images into OsiriX Dicom Viewer for initial processing and then creating 3D models using surface rendering tools. Antonia et al. [18]. used Mimics for elbow reconstruction as well as Geomagic Studio 9 (3D Systems, Morrisville, NC, USA) for further refining and mesh repairing to ensure the reasonability of elbow bone structure. Savic et al. [19] reconstructed 15 femurs using Mimics 18 (Materialize Inc., Leuven, Belgium) to establish a parameterized femur model, spending five days and achieving 97% accuracy compared to real bones. Grunert et al. [20] utilized D2P^TM^ (3D Systems Inc., Rock Hill, SC, USA) for bone segmentation, under the precision control of senior joint surgeons, to acquire the STL file for 3D printing of the elbow joint.

Overall, the recognition, segmentation, and reconstruction of the elbow joint have primarily been performed by engineers and senior surgeons. The accuracy of reconstruction was primarily influenced by the precision of segmentation, which commonly involves threshold segmentation, region-growing, pixel trimming, and supplementation. Due to the variations in bone mineral density (BMD) and the ambiguity of bone edges, ensuring segmentation precision is often challenging. Additionally, this process is time-consuming, labor-intensive, and subjective, requiring surgeons with extensive experience and specialized knowledge [21,22]. However, accurate image segmentation is paramount for precise elbow joint manipulation. To improve segmentation efficiency and accuracy, advanced image segmentation algorithms have been explored and continuously developed, especially with the emergence of convolutional neural networks (CNNs) and deep learning [23].

Image segmentation based on deep learning allows for the acquisition of complex image features and delivers accurate segmentation results in specific tasks [24]. Ronneberger et al. [25] proposed U-Net, which consists of an encoder and decoder. It has been widely applied in the field of biomedical image segmentation due to its effectiveness with small datasets. Based on U-Net, researchers improved the network structure by optimizing various components, such as the backbone network, skip connections, bottleneck structure, and residual design. These enhancements have led to the proposal of U-Net++ [26], U-Net3+ [27], AReN-UNet [28], and HAD-ResUNet [29]. Moreover, transformer architecture has been gradually applied in the field of medical image segmentation. It utilizes a mechanism known as self-attention, allowing it to effectively capture relationships between different words or elements in a sequence. Several researchers employed deep learning methods for the segmentation of elbow joints. Wei et al. [30] used Faster R-CNN to detect and locate elbow bones, and designed a global–local fusion segmentation network to solve the overlapping area identification problem. Xu et al. [31] applied PointNet++ to perform segmentation on the single-frame point cloud of the human body, enabling the generation of six parts of the human body for estimating human joints in various poses, and realizing the capture and simulation of human joint movement based on computer vision.

However, obtaining large-scale datasets is relatively difficult in some tasks, which poses a significant obstacle to training neural networks. To address this issue, foundation models have emerged and achieved remarkable progress in artificial intelligence (AI). Pre-trained on extensive datasets, foundation models often exhibit impressive generalization capabilities, showing promising performance across various downstream tasks [32]. In the field of computer vision (CV), the segment anything model (SAM) has emerged as a pivotal presence. Trained on over 11 million images, SAM serves as a foundational segmentation model, which is promptable, capable of zero-shot and few-shot generations, and demonstrates considerable potential in downstream image segmentation tasks [23,33]. However, due to significant disparities between natural and medical image fields, SAM encounters challenges in medical image segmentation [34,35]. In response, MedSAM has emerged as a foundational medical segmentation model [36]. Considering data privacy and security in medical images, MedSAM demonstrates considerable promise and importance in medical image segmentation.

The MedSAM model utilizes prompt boxes to assist in segmentation, contributing to the improvement of segmentation accuracy and efficiency. The prompt boxes provide the region of interest (ROI), guiding models to locate and segment targets more accurately. For elbow joint CT scans, multiple bones may be present in the same CT image, it is crucial to minimize interference from other bones during segmentation. Traditionally, the prompt boxes are marked manually by researchers. Undoubtedly, generating the prompt boxes automatically based on the results of bone recognition can improve automation and accuracy in bone segmentation and reconstruction. 

Therefore, a method for recognizing, segmenting, and reconstructing elbow joint bones automatically is proposed. This method can accurately and objectively identify adjacent elbow bones and generate segmentation prompt boxes, thereby achieving automation in elbow joint bone segmentation and facilitating seamless 3D reconstruction. Furthermore, this method is expected to be applied even with limited training sample sizes in clinical applications. It will provide significant assistance in clinical diagnosis, preoperative planning, and intraoperative navigation.

## 2. Method

The method can be divided into three steps, as shown in Figure 1. The elbow CT images obtained through CT scanning are used as input [33]. Firstly, a spatial elbow joint recognition is introduced, involving threshold segmentation, region-growing techniques, and elbow anatomical knowledge, and the prompt boxes are generated. Subsequently, the original CT images undergo clipping and encoding by an image encoder, obtained via transfer learning from the MedSAM. The elbow bone masks are predicted by a segmentation model, with mask correction and reclassification designed based on bone recognition results. Finally, the three main bones at the elbow joint, namely the humerus, ulna, and radius, are reconstructed and smoothed using the marching cube algorithm and Laplace smooth operation.

### 2.1. Original Elbow CT Image Segmentation

MedSAM is a typical segmentation foundational model, generated by transfer learning based on SAM [33] deploying over 11 million medical images. Functioning as a prompt segmentation method, MedSAM employs prompt boxes to specify segmentation objects, thereby reducing segmentation ambiguity. Similar to SAM, MedSAM comprises an image encoder, a prompt encoder, and a mask decoder, as illustrated in Figure 2. The image encoder maps the original CT images to a higher dimensional space, and the prompt encoder converts the prompt bounding boxes into feature representation through positional encoding. The mask decoder utilizes image embeddings, and prompt embeddings to generate masks. MedSAM demonstrates comparable or even superior segmentation accuracy to professional models, as evidenced by its performance on over 70 internal tasks and 40 external validation tasks [36]. As a foundational segmentation model, it possesses robust feature extraction capabilities due to its extensive pre-training on a large number of medical images, implying that even in the absence of specific samples or with a limited dataset, the model can leverage its internal feature representations to perform effective inference. Consequently, the MedSAM model and the MedSAM model after transfer learning exhibit powerful generalization capabilities with small sample sizes.

To improve the adaptability of MedSAM for elbow CT images, several series of CT images are utilized for transfer learning. For clarity, the fine-tuned MedSAM is referred to as elbow MedSAM. During dataset construction, several series of elbow CT images are randomly selected to generate the training dataset, and a CT image is considered as a sample. Specifically, each sample includes three parts: (1) original image, (2) image labeled by senior surgeons (binary matrix with the same shape as the original image), and (3) prompt box. The prompt box surrounding the target is automatically generated, by slightly expanding the narrow box with a random amount ranging from 0 to 30 pixels, which prevents the network from excessively relying on the boundary range of the rectangular box, which is shown in Figure 3.

Manual drawing of prompt boxes is required for MedSAM, significantly impeding the efficiency of elbow joint segmentation and reconstruction. It is crucial to automatically recognize elbow bones from original CT images to improve efficiency. An initial threshold segmentation is performed, followed by the utilization of the region-growing technique to determine the connected component labels of each voxel within the segmented foreground by applying the breadth-first search algorithm (BFS). The connected components are then sorted based on the number of voxels they encompass. Specifically, the initial threshold setting should prevent bone adhesion from occurring at the elbow joint. Additionally, the initial threshold should be set as small as possible to encompass more bone tissue during the initial segmentation process.

In standard unilateral elbow joint CT scans, the four largest connected components obtained through the initial threshold calculation correspond to the CT bed, humerus, ulna, and radius, as shown in Figure 4. Considering that the joint space is smaller than the distance between the bone and the CT bed, the minimum distances between every pair of connected components are calculated using the KD tree algorithm, and the distance matrix is recorded as follows:(1)$$D_{min}^{4\times 4}={\left\{d_{ij}\right\}}^{4\times 4},$$
where $d_{ij}$ is the minimum distance between the ith and jth connected components, and the index of CT bed can be determined as follows:(2)$$id_{CT\;bed}=\underset{i}{argmax}\left(\sum_{i=1}^{4}d_{ij}\right),$$

For the remaining three connected components, the relative distances are calculated. Several points are uniformly chosen for each connected component, and the distances from these points to other connected components are calculated. The average distance is considered as the relative distance between bones. Since the relative distance between the ulna and radius is the smallest among the three groups, the relative distance matrix $D_{mean}^{3\times 3}$ can be utilized to select the humerus from the three bones, calculated similarly to Equation (2).

The forearm shaft orientation (${\vec{n}}_{f}$) is estimated based on a vector defined by point *A* (the distal point of the humerus) and point *B* (the farthest point from point *A* in the other two connected components), as shown in Figure 5. The proximal projection location of the ulna and radius in this direction can be calculated according to the following: (3)$$p_{p}=\max\;\left(C\times {\vec{n}}_{f}\right)$$
where $C\in R^{N\times 3}$ represents the connected component of the ulna or radius. The connected component with the largest $p_{p}$ is recorded as the ulna.

The prompt box is generated based on the results of bone recognition. The voxels of connected components are projected onto each layer. A narrow prompt box can be obtained:(4)$${Box}_{narrow}=\left[\begin{array}{cccc} x_{min} & x_{max} & y_{min} & y_{max} \end{array}\right]$$

Bone recognition results cannot encompass the entire bone, it is necessary to expand the narrow prompt box with an interval d (Figure 6). Therefore, the expanded prompt box can be expressed as follows:(5)$${Box}_{\mathrm{expand}}=\left[\begin{array}{cccc} x_{min}-d & x_{max}+d & y_{min}-d & y_{max} \end{array}+d\right]$$

The expanded prompt boxes are input into the segmentation model and encoded by the prompt encoder. Finally, segmentation masks are predicted layer by layer.

### 2.2. Mask Correction and Reclassification

Several details need to be supplemented to reduce segmentation errors. Tiny holes within segmentation bones are observed, which are caused by the inhomogeneity of BMD within the bone marrow cavity. These holes should be filled using a hole-filling algorithm.

Near the elbow joint, multiple bones may appear in the same image with a narrow spacing. The mask predicted from the prompt box may not only include the target bone but also other bones. To address this issue, reclassification and validation operations are performed for the predicted mask based on the bone recognition result. Considering that a single bone might comprise multiple separated regions on a single layer, the bone reclassification result can be expressed as follows:(6)$$C=\overset{N_{C}}{\underset{i=1}{⋃}}C_{i}\cdot sign\left(\left\|C_{i}\cap C_{recognize}\right\|\right)$$
where $C_{i}$ denotes the connected component calculated based on the segmentation mask, $C_{recognize}$ denotes the point set corresponding to the bone recognition result, and $N_{c}$ indicates the number of connected components in the prompt box. Finally, the target bone can be accurately predicted, as shown in Figure 7.

### 2.3. Elbow Reconstruction

The elbow bone is reconstructed using the marching cube algorithm based on the elbow segmentation results. Then, the image acquisition coordinate system is transformed into the spatial coordinate system, facilitating precise intraoperative localization in clinical applications. Therefore, the coordinate transformation can be expressed as follows:(7)$$\left[\begin{array}{c} P^{\prime }\\ 1 \end{array}\right]=\left[\begin{array}{ccc} O_{image}\Delta S & \begin{array}{c} 0\\ 0\\ \Delta h \end{array} & S\\ \begin{array}{cc} 0 & 0 \end{array} & 0 & 1 \end{array}\right]\left[\begin{array}{c} P\\ 1 \end{array}\right]$$
where P and $P^{\prime }$ denote the vertices of the 3D bone model before and after the coordinate transformation, $O_{image}$ and ΔS represent image orientation and pixel spacing obtained from original CT data, Δh denotes slice distance, and S is the spatial position of the point in the upper left corner of the first CT image. Following that, a Laplace smoothing operation is conducted to reduce high-frequency noise while preserving the whole structure of the bone shape. During this process, the smoothed positions are determined based on their adjacent points for all vertices:(8)$$U\left(p\right)=\frac{1}{n}\sum_{i=1}^{n-1}Adj_{i}\left(P\right)$$
where P and $U\left(p\right)$ are points before and after smoothing.

## 3. Results and Discussion

### 3.1. Dataset

The study received approval from the Beijing Jishuitan Hospital Ethics Committee, and informed consent was waived by the ethics committee. A total of 41 patients treated in the Department of Trauma at Beijing Jishuitan Hospital were enrolled in the study. Inclusion criteria were no history of congenital malformations and arthritis. All patients underwent unilateral elbow CT scans (UIH/uCT500) with voltage and current settings of 100 kV and 65 mA. The resolution of the CT image was 512×512, with a pixel spacing ranging from 0.29 to 0.57 mm, and a slice distance of 0.8 mm. Among 41 series of unilateral elbow joint CT scans, there are 17 left elbow joints and 24 right elbow joints. The range of elbow flexion angles was between 82.10° and 170.11°. The entire dataset was divided into training and validation datasets, with 11 series of CT images randomly selected for the transfer learning of MedSAM, and the other 30 series of CT images applied for verification. Three senior orthopedic experts collaborated to label the bones using Mimics 20.0, including the humerus, ulna, and radius, which were regarded as the “ground truth (GT)”.

The transfer learning of MedSAM was conducted using Python 3.10 and PyTorch 2.0. The checkpoint of MedSAM is used as the initial value. Adam optimizer was utilized to train the network; the training spanned 100 epochs, utilizing a weight decay of 0.01, and a learning rate set of 0.0001. The loss function is defined by the sum of cross-entropy loss and dice loss, which is reduced from 0.022 to 0.007 for the training dataset. 

### 3.2. Verification of Elbow Bone Recognition

#### 3.2.1. Result of Elbow Bone Recognition

Figure 8 shows the bone recognition results and generated prompt boxes, demonstrating that elbow bone recognition can automatically identify multiple targets in the entire CT images based on the bone position relationships. This facilitated more efficient and intuitive classification of bones in elbow CT images. However, the recognized masks only depicted the general shape of the bones. Specifically, only pixels with high CT intensity values can be captured. The marrow cavity was omitted due to the low bone mineral de BMD, which can be supplemented if the cortical bone is completely segmented. In addition, the BMD at the bone end is relatively low due to the need for flexibility and shock absorption, resulting in missing pixels, which typically require pixel-by-pixel segmentation refinement. Despite these limitations, the recognized masks still provided an approximate indication of the bone’s position, guiding the generation of prompt boxes. It helps enhance the automation of subsequent segmentation and 3D reconstruction. To ensure complete segmentation of elbow bones, expanding the prompt boxes is conducted. In our studies, the expansion size was set to 20 pixels at the bone ends and 10 pixels at the bone shafts.

#### 3.2.2. Impact of Automatic Prompt Box Generation

An ablation study was performed to verify the significance of prompt box generation, which is shown in Figure 9. Obviously, without a prompt box, the segmented mask covered almost the entire image, as shown in Figure 9b, resulting in an invalid segmentation. In addition, for an inappropriate prompt box (such as an overly large prompt box, as shown in Figure 9c), effective segmentation results could not be obtained. This indicates that appropriate prompt boxes are crucial for achieving precise segmentation.

Furthermore, it is crucial to recognize elbow bones and generate prompt boxes automatically, as manually drawing prompt boxes by surgeons is time-consuming and labor-intensive. Additionally, identifying the specific elbow joint bone within the prompt box is challenging when only a single image is available, particularly in scenarios where one bone appears as two separate domains in the CT image.

### 3.3. Impact of Mask Correction and Reclassification

The comparison for a series of CT images before and after mask correction and reclassification was performed, and the intersection over union (IoU) was utilized to gauge segmentation accuracy, which is defined by the following:(9)$$IoU=\frac{A\cap B}{A\cup B}$$
where A and B represent two masks being compared, and the numerator means the area of the overlap regions of two masks, and the denominator means the total coverage of two masks, as shown in Figure 10. Obviously, IoU is 1 if the two masks completely overlap, while IoU is 0 if the two masks are entirely unrelated.

It was noticeable that mask correction had little impact on the IoU at the bone shaft (Figure 11a). The slight differences in IoU values for these layers were primarily due to minor holes attributed to the complex shape and texture of bone tissue. The IoU values were improved after these tiny holes were filled, as shown in Figure 11b.

However, in the region of the humeral trochlea and capitellum, the difference in IoU values was substantial, with a maximum difference of 0.228 (Figure 11c). Due to the narrow joint space at the humeroulnar joint, there was an overlap between the prompt boxes for the ulna and humerus. As a result, segmented masks within the humerus prompt box might not entirely belong to the humerus (Figure 11c). However, through mask reclassification based on the initial bone recognition, pixels corresponding to the ulna were excluded, resulting in a significant increase in IoU, highlighting the importance of reclassifying segmentation targets.

### 3.4. Result of Elbow CT Segmentation

#### 3.4.1. Qualitative Evaluation of Segmentation Results

Figure 12 demonstrates the qualitative evaluation of segmentation results using U-Net, origin MedSAM, and elbow MedSAM. There was no significant difference observed for the bone shafts of the humerus (Image 1), ulna, and radius (Image 2) among the three segmentation models. However, the primary segmentation challenge arose near the elbow joint, primarily due to the narrow joint space between the humerus and ulna, resulting in an indistinct boundary between bone and soft tissue. For U-net, it is almost impossible to separate different bones. In the region of the elbow joint, different bones might be segmented as an entire bone (Image 4). Among three segmentation methods, elbow MedSAM demonstrated excellent segmentation results (Images 3 and 4), attributed to the transfer learning, which enhanced the model’s effectiveness in distinguishing between joint gaps and bone tissues. Obviously, transfer learning resulted in a sharper feature recognition capability. Even when confronted with small joint spaces, elbow MedSAM demonstrated a reduced likelihood of identifying different bones as a single target, thereby ensuring the relative integrity of bone segmentation. Furthermore, our investigation revealed that even in cases where the elbow joint is in a flexed posture, meaning elbow bones are not fully displayed in cross-section, elbow MedSAM can accurately identify and segment the forearm bones (Image 5).

#### 3.4.2. Quantitative Evaluation of Segmentation Accuracy

The U-Net exhibited extremely limited segmentation performance at the elbow joint regions. Moreover, it has also been documented in the literature that its segmentation effectiveness is inferior to that of MedSAM for medical images [36]. Therefore, only the origin MedSAM and elbow MedSAM were compared for quantitative evaluation.

The quantitative evaluation was first conducted on a series of CT scans, as shown in Figure 13. It was clear that the elbow MedSAM exhibited stability, with the IoU values ranging from 0.911 to 0.983, 0.894 to 0.981, and 0.865 to 0.980 for the humerus, ulna, and radius, respectively. Compared to the origin MedSAM, the IoU values were significantly improved.

In fact, the bone shaft exhibits high BMD and clear bone texture, resulting in relatively good segmentation results for both models. However, near the elbow joint, compared to elbow MedSAM, the origin MedSAM experienced a maximum decrease in IoU values of 0.31, 0.138, and 0.38 for the humerus, ulna, and radius, respectively.

The CT images displaying the most significant disparities in IoU values were utilized for further analysis. For the humerus and ulna, origin MedSAM exhibited the lowest segmentation accuracy at the humeroulnar joint, mainly attributed to the extremely narrow joint space and blurred edges, as shown in Figure 14. Without transfer learning, the segmentation model might erroneously identify the humerus and ulna as a single bone, and the accuracy was hardly enhanced through mask reclassification. For the radius, lower segmentation accuracy was observed near the radial head due to physiological reasons. The radial head requires flexibility and cushioning during elbow joint movement, thus resulting in relatively lower BMD. Additionally, the BMD naturally decreases with increasing age.

Lower IoU values mainly existed in several layers near the bone end. However, the phenomenon did not indicate poor segmentation quality. In these regions of the CT images, the target bones occupied a very small proportion (just a few dozen pixels). As a result, even a small number of pixels inconsistent with the ground truth can lead to a significantly low IoU value. Moreover, influenced by the partial volume effect, the bone edges appeared quite blurred. This implies that even senior surgeons might not provide completely accurate segmentation results when labeling the ground truth. Furthermore, we noted that despite the partial volume effect causing blurred bone edges, our model demonstrated a certain degree of robustness. Overall, the results indicated that despite encountering some challenges, the elbow MedSAM has achieved satisfactory results in elbow joint CT image segmentation tasks.

Statistics analysis was conducted on the segmentation results among the entire test dataset, and the results were visualized in a box plot, as depicted in Figure 15. The boxes encompassed IoU values from 10% to 90%. From the figure, the median IoU values were 0.963, 0.959, and 0.950 for the humerus, ulna, and radius, respectively. For 90% of the elbow CT images, the segmentation IoU values exceeded 0.939, 0.927, and 0.917, significantly higher than the IoU values achieved by origin MedSAM, which were 0.77, 0.773, and 0.779, respectively. This indicated that elbow MedSAM demonstrated higher stability and can be applied to a broader range of elbow CT images. In contrast, the origin MedSAM showed weaker specificity toward the elbow joint, resulting in slightly inferior performance.

#### 3.4.3. Objectivity Discussion of Elbow Joint Segmentation

In clinical practice, the segmentation results of elbow joint bones by senior surgeons exhibit a certain degree of subjective judgment, highlighting the reliance on the clinician’s experience for manual segmentation. This subjectivity can lead to inconsistencies and variability in the results. To address this issue, we developed Elbow-MedSAM through transfer learning based on MedSAM, which has been pre-trained on a large dataset of medical images. This model leverages the extensive segmentation experience embedded in these images, thereby eliminating the need for manual adjustments and human intervention. By automating the segmentation process, Elbow-MedSAM ensures a more objective and consistent approach, reducing the impact of individual biases and potentially improving the accuracy and reproducibility of the segmentation outcomes.

### 3.5. Reliability and Accuracy of 3D Elbow Reconstruction

#### 3.5.1. Result of 3D Elbow Reconstruction

The comparison between the 3D elbow bone surface models reconstructed by the automatic method and ground truth was conducted, and the reconstruction results are depicted in Figure 16. It is evident that the 3D elbow joint model obtained through the automatic method is essentially consistent with the ground truth, showing no obvious holes or bone adhesions. 

The surface error of the reconstructed model was calculated, with the statistical results shown in Figure 17. The maximum surface error values were 0.877 mm, 1.246 mm, and 1.274 mm for the three elbow bones. These results underscored the reliability and accuracy of our automatic reconstruction method in capturing the intricate details of the elbow joint morphology, indicating a high degree of consistency with the ground truth.

#### 3.5.2. Reliability Analysis of Elbow Joint Reconstruction

Further analysis of the reconstruction results of 30 elbow joints was conducted, and the maximum surface errors are shown in Table 1 and Figure 18. Two main reasons accounted for the occurrence of large surface errors at individual points: (1) incomplete segmentation near the radius head in some cases, as depicted in Figure 18a. (2) For the bone shaft positioned at the edge of the CT image, the edges of the rectangular box coincide with the image edge. This situation primarily occurred in the ulnar and radial shaft regions, resulting in part bone marrow cavities in the reconstructed bone models, introducing surface errors from the ideally reconstructed bones, as shown in Figure 18b. Since the main functions of the elbow joint are flexion and extension, these errors are not expected to significantly affect the analysis of elbow joint morphology and kinematic simulation.

The automatic reconstruction was successfully implemented in all 30 cases of elbow joint CT images, indicating its robustness and general applicability for elbow joint CT images. As the checkpoint is fixed after transfer learning, the segmentation results are repeatable. The reconstructed surface models generally align well with the ground truth except for some special positions, and the reconstruction error is within an acceptable range, further validating the reliability of the proposed method.

## 4. Conclusions

In this study, an automatic method for elbow joint recognition, segmentation, and reconstruction is proposed. Prompt boxes are generated by automatic elbow bone recognition. Transfer learning is utilized to improve the segmentation accuracy, and hole-filling and mask reclassification are applied to refine the segmented masks. The elbow bone reconstruction can be conducted seamlessly. By comparing the segmentation and reconstruction results with manually labeled images, this automatic method has proved to be reliable, objective, and accurate. The main conclusions are as follows: (1)This study employs an interpretable algorithm to automatically recognize the humerus, ulna, and radius from elbow joint CT images. The algorithm exhibits stability and effectiveness for elbow joints from flexion (82.10°) to extension (170.11°) postures.(2)The IoU values near the joint are significantly increased by mask correction and reclassification, with a maximum improvement of 0.028, conclusively boosting segmentation accuracy.(3)The segmentation accuracy is enhanced by the MedSAM after transfer learning, allowing for more precise capture of bone edges and reducing instances of mistaking multiple bones as a single target. The median IoU values are 0.963, 0.959, and 0.950 for the humerus, ulna, and radius, respectively, notably surpassing the predictions of the origin MedSAM.(4)The maximum surface errors for the bone surface model reconstructed by the marching cube algorithm are 1.127, 1.523, and 2.062 mm for the humerus, ulna, and radius, respectively.

## Figures and Tables

**Figure 1 sensors-24-04330-f001:**
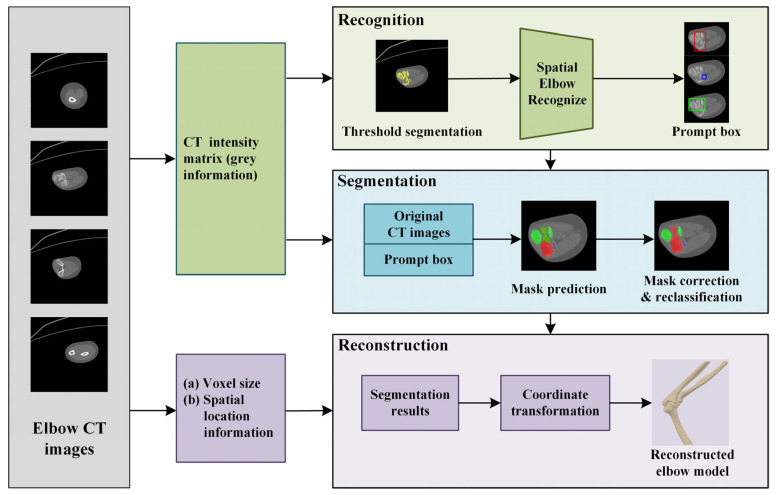
The elbow recognition, segmentation, and reconstruction method. In figure, the humerus, ulna and radius are shown in red, green, and blue, respectively.

**Figure 2 sensors-24-04330-f002:**
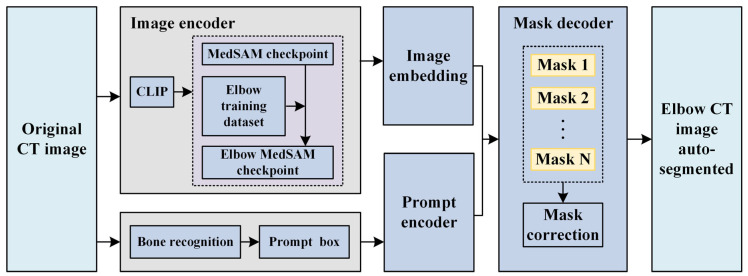
Automatic elbow CT image segmentation.

**Figure 3 sensors-24-04330-f003:**
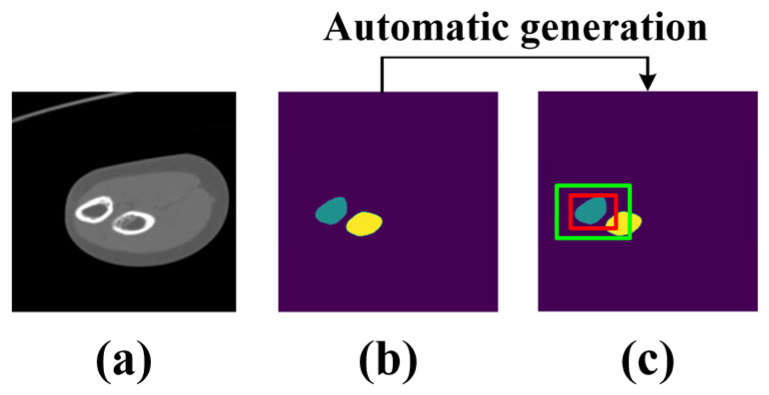
Composition of the training dataset. (**a**) Original image; (**b**) labeled image; (**c**) labeled image with automatically generated prompt box. The red and green rectangular boxes are narrow box and expanded box, respectively.

**Figure 4 sensors-24-04330-f004:**
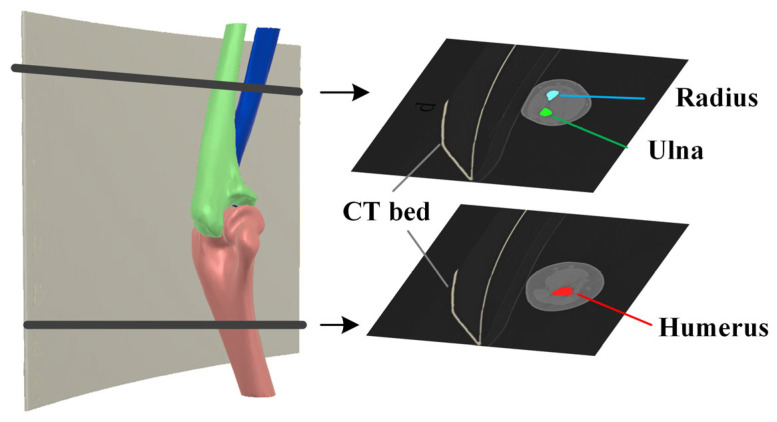
Main connected components after threshold segmentation and region-growing.

**Figure 5 sensors-24-04330-f005:**
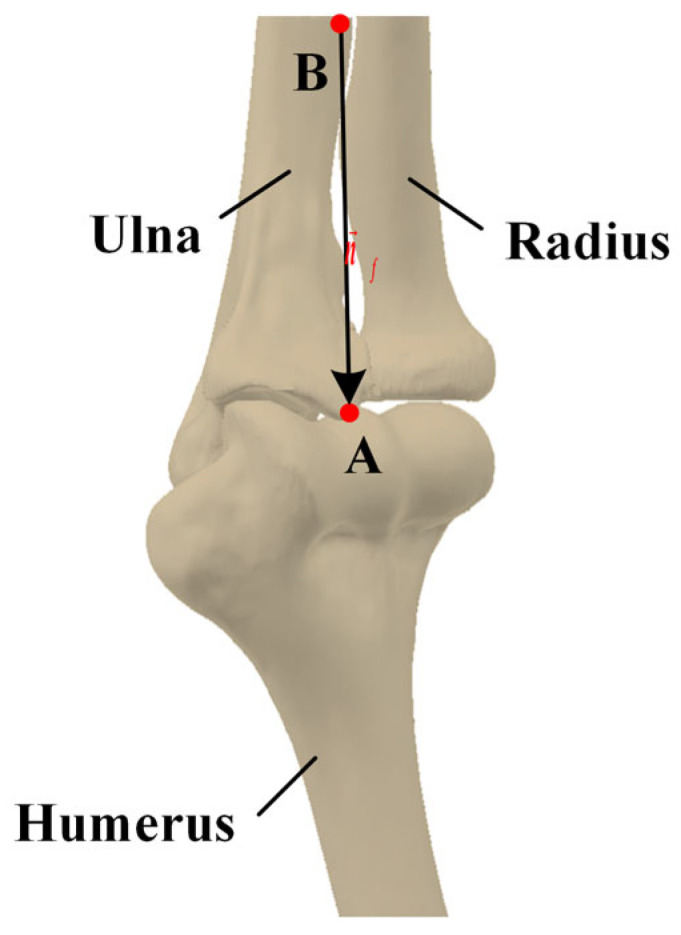
Evaluation of forearm shaft orientation.

**Figure 6 sensors-24-04330-f006:**
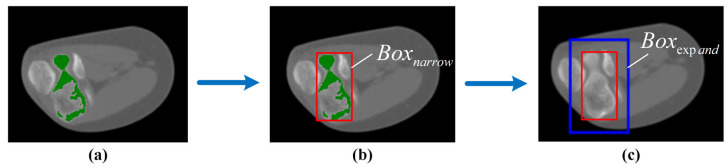
Automatic prompt box generation and expansion. (**a**) Bone recognition; (**b**) narrow prompt box; (**c**) expanded prompt box.

**Figure 7 sensors-24-04330-f007:**
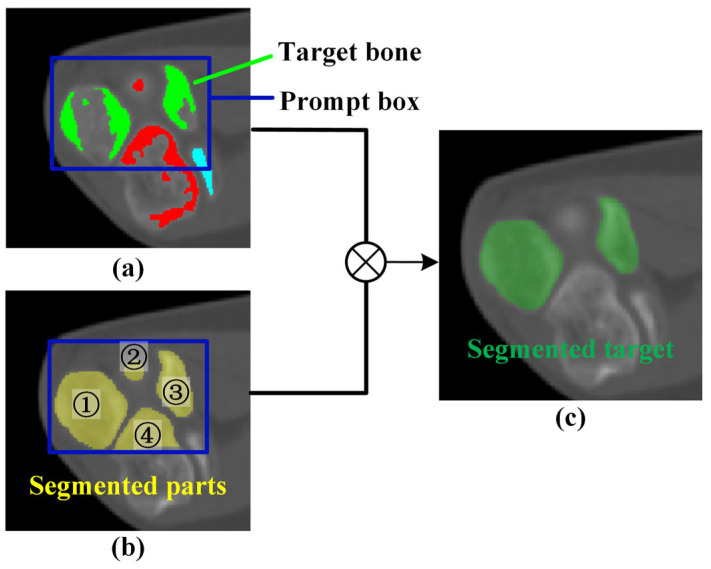
Schematic of mask reclassification. (**a**) Result of bone recognition; (**b**) automated segmentation; (**c**) mask after reclassification.

**Figure 8 sensors-24-04330-f008:**
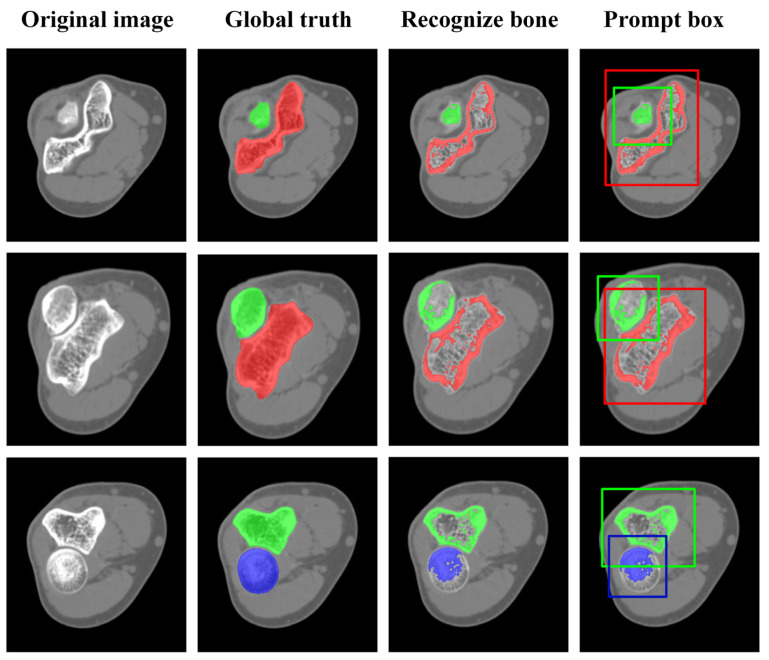
Results of elbow bone recognition and generated prompt boxes. In the figure, the humerus, ulna, and radius are shown in red, green, and blue, corresponding to the prompt boxes.

**Figure 9 sensors-24-04330-f009:**
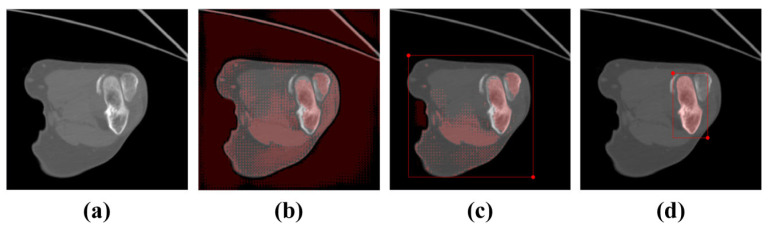
The contrasting impact of prompt boxes on elbow bone segmentation. (**a**) Original image; (**b**) segmentation masks without a prompt box; (**c**) segmentation masks using an inappropriate prompt box; (**d**) segmentation masks using an appropriate prompt box.

**Figure 10 sensors-24-04330-f010:**
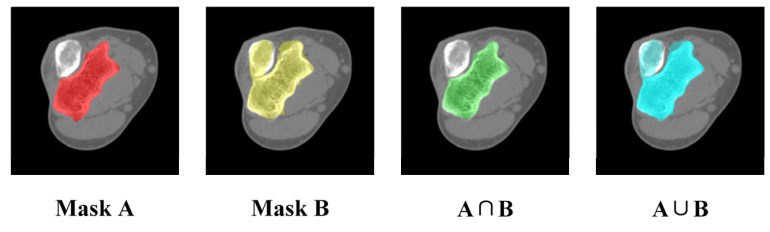
Diagram about the calculation of IoU.

**Figure 11 sensors-24-04330-f011:**
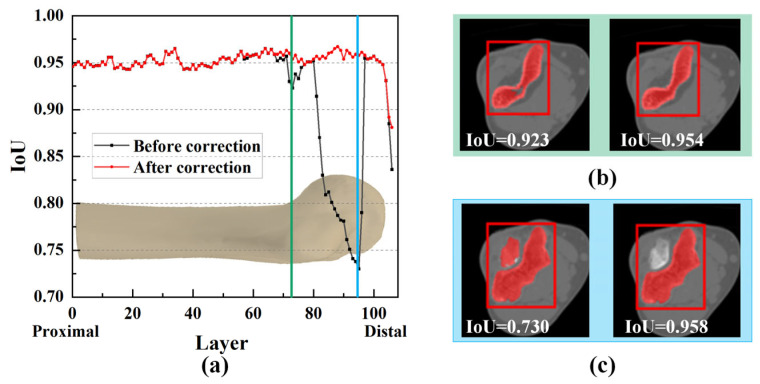
Comparison of the effect of mask correction and reclassification. (**a**) Comparison of IoU values for all CT layers; (**b**,**c**) mask comparison for two specified layers.

**Figure 12 sensors-24-04330-f012:**
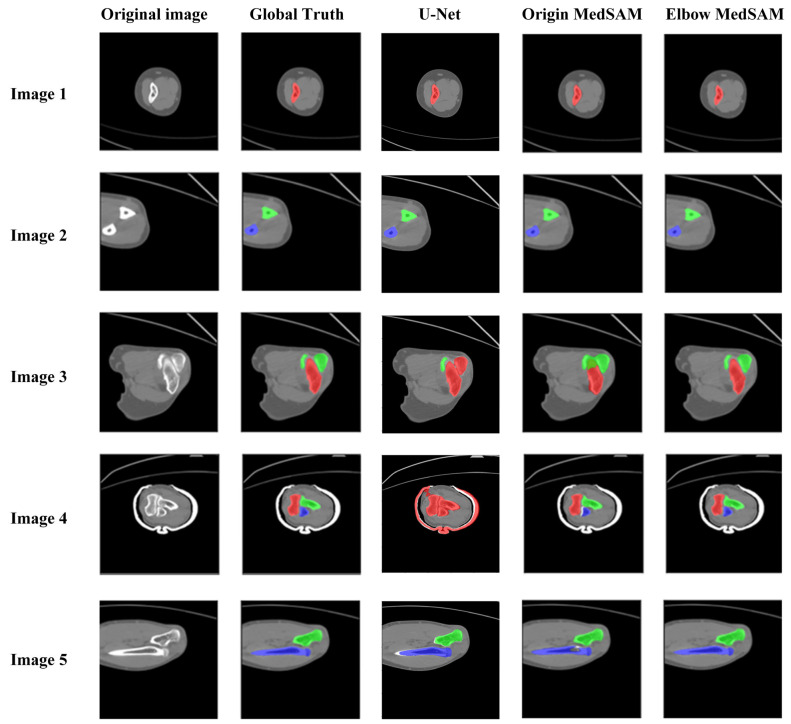
Comparison of segmentation results between origin MedSAM and elbow MedSAM. In the figure, the humerus, ulna, and radius are shown in red, green, and blue, respectively.

**Figure 13 sensors-24-04330-f013:**
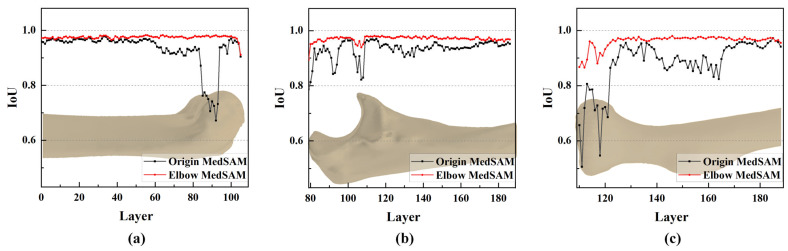
IoU plot of a series of elbow joint CT segmentation results. (**a**) Humerus; (**b**) ulna; (**c**) radius.

**Figure 14 sensors-24-04330-f014:**
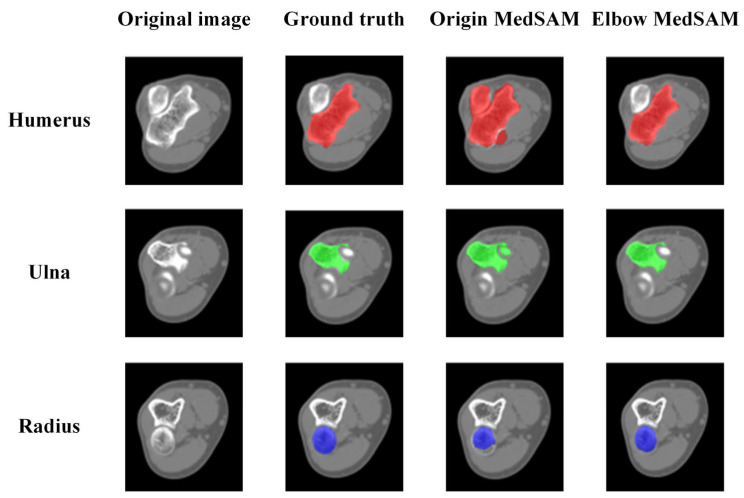
Instances of segmentation result for each bone. In the figure, the humerus, ulna, and radius are shown in red, green, and blue, respectively.

**Figure 15 sensors-24-04330-f015:**
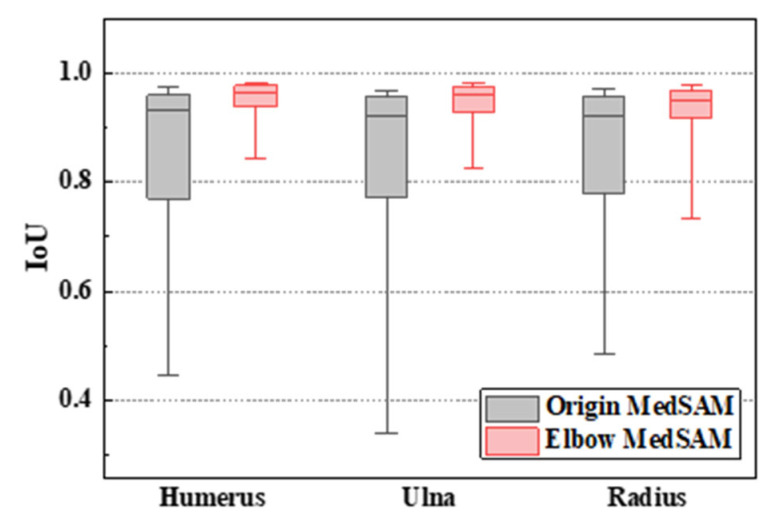
Boxplot of segmentation IoU values of elbow CT images.

**Figure 16 sensors-24-04330-f016:**
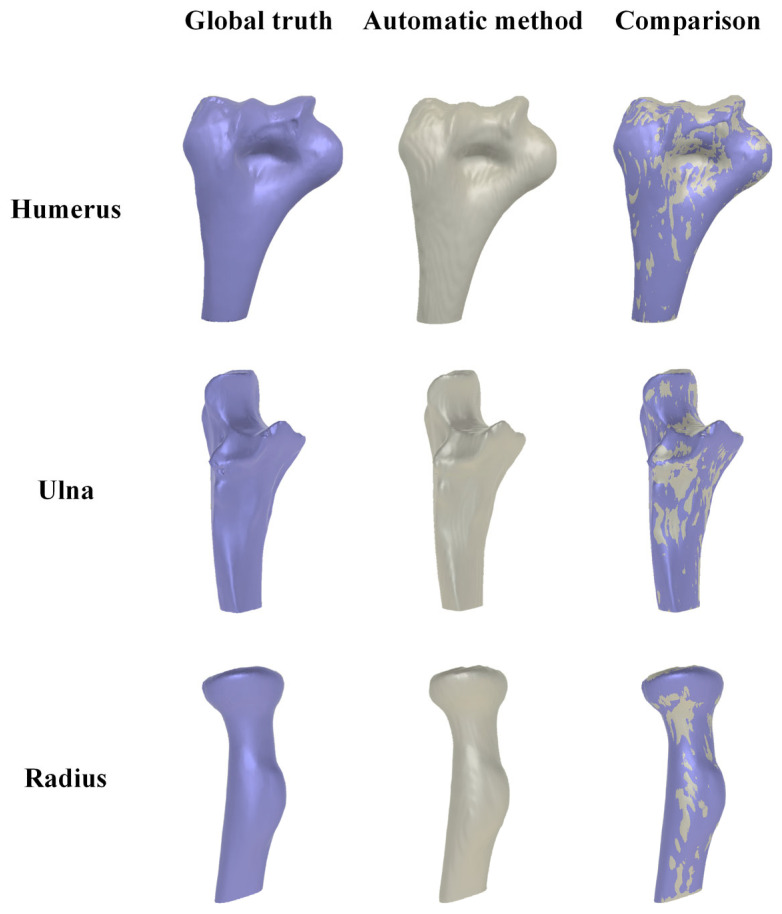
Comparison of reconstruction elbow bone model between ground truth (in purple) and automatic method (in grey).

**Figure 17 sensors-24-04330-f017:**
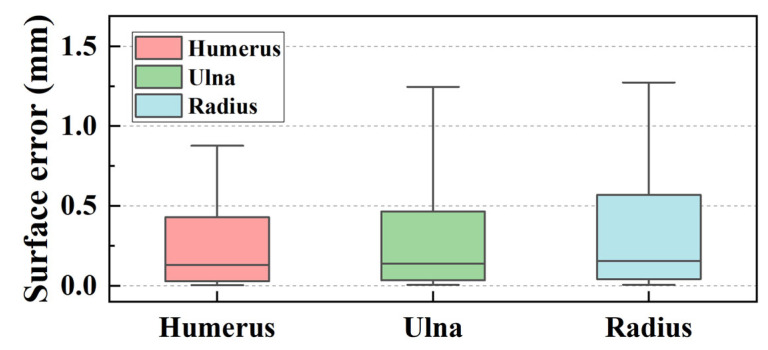
Boxplot of surface error values of reconstructed models between automatic method and ground truth.

**Figure 18 sensors-24-04330-f018:**
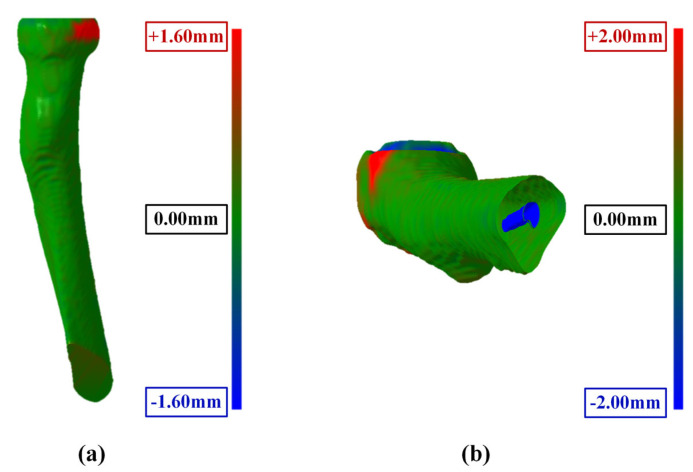
Main error in surface model reconstruction. (**a**) Incomplete segmentation near the radius head; (**b**) interference of marrow cavity.

**Table 1 sensors-24-04330-t001:** Max surface error for test CT series.

	Mean (mm)	Quartiles (Q1 to Q3) (mm)	Range (Min. to Max.) (mm)
Humerus	1.127	0.654 to 1.433	0.262 to 2.247
Ulna	1.523	0.976 to 1.906	0.737 to 2.695
Radius	2.062	1.299 to 2.711	0.582 to 3.388

## Data Availability

The data used to support the findings of this study are available from the corresponding author upon request. The data are not publicly available due to privacy concerns. We are actively working with hospitals to obtain public permission for the dataset.

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
