# Peer review of "An Automatic Method for Elbow Joint Recognition, Segmentation and Reconstruction"

_sensors, 2024, doi:10.3390/s24134330_

Round 1
Reviewer 1 Report
Comments and Suggestions for Authors
This manuscript presents a new method for the detection, segmentation, and reconstruction of the elbow joint from CT scans. The method seems to be an improvement over an existing approach, MedSAM. The new approach is relatively well described. Few issues are identified in the way the method was tested and the conclusions. Specifically:
page 2, line 93 - what supports the claim that the method will work with limited training sets? There is nothing in the text that shows this.
page 7 - Dataset - Unclear description. Issues (a) Are all 41 bilateral CT scans of normal elbows? (b) Are they used independently, i.e., 82? (c) How do the 17 extra CT elbow scans used in this study? Are they added to the 82? Is there a distinction between left and right? And the total number of scans is 99 from which 11 are used for training? Very confusing, please clarify.
page 8, line 17 - I assume by "global truth" you mean "ground truth". Please change to ground truth all related references in the text. Global truth is not used in medical imaging unless you mean something else, in which case you need to clarify.
page 8, equation 8 - you should define what is A and B in this equation preferably with reference to figure 8.
page 14 - line 368 - the claim about the accuracy of the proposed method is reasonable and is supported by the comparison of the results to the ground truth. However, how do you conclude that the method is reliable and objective? What does objective mean in this application? What tests did you do to demonstrate these?
Minor issues:
page 2, line 91 - "generate" instead of "generating"
page 5, line 157 - error in reference source (there are a couple of other places too in the text where references are missing)
page 8, figure 7. I think is should be "Original image" instead of "Origin image" because this does not refer to the "origin MedSAM".
page 10, line 283 - "Quantitative evaluation" instead of "Quantitative"
page 14, line 368 - "accurate" instead of "accuracy"
Comments on the Quality of English Languageminor issues/see a few listed above
Reviewer 2 Report
Comments and Suggestions for Authors
In this paper, the author proposed a method to segment elbow joint automatically.
The idea is to adopt the Medical Segment Anything foundation network with a prompt box recognition, which involving threshold segmentation, region-growing techniques and elbow anatomical knowledge, and the prompt boxes are generated. Then, the original CT images undergo clipping and encoding by image encoder. The MedSAM was finetuned with Transfer learning technique. The elbow bone masks are the predicted by the finetuned segmentation model. Mask correction and reclassification is achieved with the segmentation result and the bone recognition at the first stage. The three main bones at the elbow joint, namely the humerus, ulna and radius, are then reconstructed and smoothed using Marching Cube algorithm and Laplace smooth operation.
The works reported in the paper is novel. Results show that the proposed technique is superior than SAM. However, it lacks some of the details. Here are specific comments:
- Transfer learning details is missing. How is the own data set for finetuning? There is no description nor any sample data with labels available.
- There are some other existing methods for the elbow joint segmentation, such as,
Tianxu Xu, Dong An, Yuetong Jia, Jiaqing Chen, Hongkun Zhong, Yishen Ji, Yushi Wang, Zhonghan Wang, Qiang Wang, Zhongqi Pan, Yang Yue, "3D joints estimation of human body using part segmentation," Information Sciences, Volume 603, 2022, Pages 1-15.
- The author should cite more state-of-the-art segmentation algorithm.
- Please compare more other segmentation methods.
- The automatic elbow location recognition directly affect the success and performance of segmentation. Please perform ablation study to compare to the baseline which without automatic prompt generation.
- Line 157, 347: type setting error "Equation Error! Reference source not found..", please correct it.
- Line 473: Reference 30 is not a complete citation. Please revise. Check it from https://doi.org/10.48550/arXiv.2305.08196
Comments on the Quality of English Language
NIL
Author Response
Please see the attachment。

Round 2
Reviewer 2 Report
Comments and Suggestions for Authors
In the revised manuscript, the authors added more detailed description of transfer learning; compared more other segmentation methods; and corrected many typesetting and citation errors.